# Intercalation-type catalyst for non-aqueous room temperature sodium-sulfur batteries

Jiarui He [1], Amruth Bhargav [1], Laisuo Su [1], Harry Charalambous[2] &
Arumugam Manthiram [1] ✉

Ambient-temperature sodium-sulfur (Na-S) batteries are potential attractive alternatives to lithium-ion batteries owing to their high theoretical specific energy of 1,274 Wh kg$^{-1}$ based on the mass of Na$_2$S and abundant sulfur resources. However, their practical viability is impeded by sodium polysulfide shuttling. Here, we report an intercalation-conversion hybrid positive electrode material by coupling the intercalation-type catalyst, MoTe$_2$, with the conversion-type active material, sulfur. In addition, MoTe$_2$ nanosheets vertically grown on graphene flakes offer abundant active catalytic sites, further boosting the catalytic activity for sulfur redox. When used as a composite positive electrode and assembled in a coin cell with excess Na, a discharge capacity of 1,081 mA h g$_s^{-1}$ based on the mass of S with a capacity fade rate of 0.05% per cycle over 350 cycles at 0.1 C rate in a voltage range of 0.8 to 2.8 V is realized under a high sulfur loading of 3.5 mg cm$^{-2}$ and a lean electrolyte condition with an electrolyte-to-sulfur ratio of 7 µL mg$^{-1}$. A fundamental understanding of the electrocatalysis of MoTe$_2$ is further revealed by in-situ synchrotron-based operando X-ray diffraction and ex-situ time-of-flight secondary ion mass spectrometry.

Room temperature (RT) sodium-sulfur (Na-S) batteries emerge as strong contenders for the next-generation energy storage systems. This recognition stems from their favorable sustainability and economic attributes, owing to their cost-effectiveness and the abundance of both sodium and sulfur in the Earth's crust[1–6]. Moreover, Na-S batteries have high theoretical specific capacity of 1675 mA h g$_s^{-1}$ (g$_s$ refers to based on the mass sulfur) and specific energy of 1274 Wh kg$^{-1}$ (based on the mass of Na$_2$S), which are optimal for energy storage systems[7–9]. However, several technical challenges remain before the practical applications of Na-S batteries. The poor electronic conductivity of sulfur (5 × 10$^{-28}$ S m$^{-1}$ at 25 °C) limits electron transport in the cathode, resulting in low sulfur utilization[10–13]. Moreover, the Na polysulfide (NaPSs) intermediates are highly soluble in organic electrolytes causing the notorious shuttle effect, which leads to substantial active material loss and interfacial deterioration[14–16].

Inspired by the developments of lithium-sulfur batteries, including the hybrid strategies of TiS$_2$, Mo$_6$S$_8$, and S[17,18], intensive research efforts have been focused in the Na-S field, such as the combination of conductive carbon and sulfur, modification of separators with carbon, and development of sulfides cathode[19–26]. These systems have illustrated decent electrochemical performance, but only with a low sulfur loading (<2 mg cm$^{-2}$). The lifespan of the Na-S batteries with high sulfur loadings (>4 mg cm$^{-2}$) still needs significant improvement. In addition, an excessive amount of electrolyte is required in these systems to guarantee a complete conversion from S to Na$_2$S in Na-S batteries. Recently, the catalytic effect was demonstrated to be an efficient way to promote the transformation of NaPSs, improving the redox kinetics in Na-S batteries[27,28]. For example, Zhang et al. prepared Mo$_2$N-W$_2$N heterostructures as a catalyst for RT Na-S batteries to accelerate the kinetics of the redox[29], exhibiting a stable cycling performance over 400 cycles with a sulfur loading of 2.7 mg cm$^{-2}$. However, recently reported work is mainly based on carbonate electrolytes, where the flooded electrolyte amount is required to thoroughly wet the glass fiber separator. He et al. developed a localized high-concentration

[1]Materials Science and Engineering Program and Texas Materials Institute, The University of Texas at Austin, Austin, TX 78712, USA. [2]X-ray Science Division, Advanced Photon Source, Argonne National Laboratory, 9700 S. Cass Avenue, Argonne Lemont, IL 60439, USA. ✉e-mail: rmanth@mail.utexas.edu

electrolyte (LHCE) that enables a stable Na-S battery under a decent amount of electrolyte condition[30]. But the capacity is only 675 mA h $g_s^{-1}$ after 300 cycles with a sulfur loading of 2 mg cm$^{-2}$ at a low rate of 0.1 C (1 C = 1675 mA $g_s^{-1}$), which is far from the theoretical capacity of 1675 mA h $g_s^{-1}$ of Na-S batteries. Recently, several sulfide electrocatalysts, including $MoS_2$, $ZnS$, $FeS_2$, and $ZnS\text{-}CoS_2$, have been shown to have an intercalative behavior which can further promote the kinetics of the redox both in Li-S and Na-S batteries[31–36]. All of those work further motivates us to develop a new electrocatalysts to improve the electrochemical performance with high sulfur loading. In this regard, achieving high-performance Na-S batteries under rigorous conditions, including high active material loading (>3 mg cm$^{-2}$) and lean electrolyte (electrolyte-to-sulfur ratio: <15 µl mg$^{-1}$), is still challenging, which is crucial to achieving the practical application of Na-S batteries. Therefore, the utilization of sulfur and the kinetics of the redox need to be further improved.

Here, we propose a concept of the intercalation-conversion hybrid cathode to promote the kinetics of sulfur redox and improve the utilization of sulfur by introducing electrochemically active $MoTe_2$ with fast sodium intercalation reaction into the sulfur cathode for Na-S batteries. As one of the many transition-metal dichalcogenides (TMDs), $MoTe_2$ possesses a much wider interlayer distance of 0.69 nm than $MoS_2$ (0.61 nm)[37]. The large interlayer space can easily accommodate the large Na-ions, resulting in better TMDs structural stability during the cycling process. Moreover, the electronic conductivity of $MoTe_2$ is 1.8 S cm$^{-1}$ (25 °C), which is nine times higher than that of $MoSe_2$ (0.2 S cm$^{-1}$, 25 °C)[38]. Such a result demonstrates that the $MoTe_2$ can further promote the transport of electrons. $MoTe_2$ in the voltage range of 0.8 to 2.8 V (vs. Na$^+$/Na) can convert to $Na_xMoTe_2$. This conversion is intercalative and thus has faster kinetics[39], which is similar to the prior studies for Li-S batteries as mentioned above where sulfide electrocatalysts were reported to exhibit intercalation. In addition, we rationally design the $MoTe_2$ with a nanosheet morphology which vertically grows on the graphene flakes. Such a nanostructure provides abundant active catalytic sites, further boosting catalytic activity for conversion between S and $Na_2S$ in Na-S batteries. The catalytic mechanism with rich active catalytic sites of hybrid cathode enables stable cycling performance with a low capacity fade rate of 0.02% per cycle after 900 cycles with practical parameters including lean electrolyte and high sulfur loading.

## Results

### Fabrication and characterization of the MTG host

$MoTe_2$ nanosheets were rationally prepared by a facile hydrothermal reaction of $MoO_3$, Te powder, and hydrazine hydrate at 220 °C for 24 h to obtain an intercalation-conversion hybrid cathode. Graphene oxide was introduced into the above reaction to improve the catalytic activity sites. Owing to the rich functional group on the graphene oxide providing abundant anchor sites, the $MoTe_2$ can uniformly grow on the graphene sheet, forming a $MoTe_2$-graphene sheet (MTG). More importantly, the architecture with a few-layered $MoTe_2$ nanosheets that are vertically aligned on graphene surfaces was obtained through the self-assembly of $MoTe_2$ nanosheets on graphene (Fig. 1a). Such an architecture ensures more abundant exposed active edge sites of $MoTe_2$ nanosheets to encourage a fast Na$^+$ intercalation. As shown in Fig. 1b, when the MTG is employed as a catalyst for Na-S batteries, it shows several advantages. First, the intercalation-type $MoTe_2$ can provide fast Na-ion transport, ensuring fast reaction kinetics during charge-discharge cycling. Second, the architecture of vertically aligned few-layered $MoTe_2$ on graphene provides more abundant exposed active edge sites for catalyzing sulfur species redox. Third, the highly conductive graphene skeleton guarantees fast charge transfer, further accelerating the sulfur species conversion reaction.

The scanning electron microscopy (SEM) images in Fig. 2a–c reveal that $MoTe_2$ nanosheets (<10 layers, as shown in Fig. 2e) are

homogeneously and vertically aligned on the surface of the graphene nanoflakes, forming a heterostructured micro forest. The transmission electron microscopy (TEM) image in Fig. 2d shows that the few-layered $MoTe_2$ nanosheets grow on both sides of the graphene flakes since both are accessible to the $MoTe_2$ precursors during the reaction. Such an architecture further improves the exposure of active sites, offering rich electrode/electrolyte interfaces for sulfur conversion reactions and efficient ionic diffusion. The high-resolution TEM image in Fig. 2e indicates the interlayer spacing of 0.73 nm, corresponding to the (002) plane of $MoTe_2$. It should be noted that the interlayer spacing is larger than that of bulk $MoTe_2$ (JCPDS No. 071-2157), which has an interlayer spacing of 0.69 nm. The energy-dispersive X-ray spectroscopy (EDAX) elemental mappings of the MTG in Fig. 2f demonstrate that Mo and Te are uniformly distributed on the C region, evidencing that the ultrathin $MoTe_2$ nanosheets are evenly aligned on the graphene flakes.

The X-ray diffraction (XRD) pattern in Fig. 2g can be indexed to be the hexagonal phase of 1 T′-$MoTe_2$. Compared to the standard XRD pattern (JCPDS No. 071-2157), the (002) peak displays a negative shift by 0.5°. The d-spacing of the (002) plane can be calculated to be 0.73 nm, which matches the TEM data well. The Brunauer–Emmett–Teller analysis in Fig. 2h further indicates that MTG possesses a porous structure, offering rich loading sites for sulfur. The corresponding pore size distribution via the Barrett–Joyner–Halenda (BJH) method is given in Fig. 2i, which indicates the bimodal mesopores in MTG.

### Interaction between MTG and sulfur species

In order to understand the underlying mechanism of MTG in promoting kinetics of sulfur redox reaction for Na-S batteries, the MTG/S and C/S cathodes with high sulfur loading of 3.5 mg cm$^{-2}$ were tested in the localized high-concentration electrolyte (LHCE) as reported in our previous work[30]. In detail, the LHCE consists of sodium bis(fluorosulfonyl)imide (NaFSI), 1,2-Dimethoxyethane (DME), and 1,1,2,2-tetrafluoroethyl 2,2,3,3-tetrafluoropropyl ether (TTE) with an molar ratio of 1:1.2:1. The LHCE can not only mitigate the NaPSs shuttling, but can also protect the Na anode from dendritic growth. As shown in Fig. S1, the Na||Cu cell with LHCE at 1 mA cm$^{-2}$ with an areal capacity of 1 mAh cm$^{-2}$ can deliver a stabilized Coulombic efficiency of 95.2% at 50 cycles. The discharge profile of the C/S cathode in Fig. 3a consists of a plateau followed by a slope, which is different from the single-sloping profile in our previous report[30]. This plateau at 2.1 V in the discharge curve is mainly attributed to the NaPS formation. Although the LHCE can significantly mitigate the NaPS dissolution into the electrolyte, the C/S cathode with higher sulfur loading will form a lot of NaPSs during the conversion, which is beyond the ability of LHCE to suppress NaPSs dissolution. Therefore, an obvious plateau related to the NaPS formation can easily be recognized. In addition, a larger voltage hysteresis ($\Delta E$, voltage difference between the charge and discharge profiles at the 50% capacity of the cells) of 0.97 V was detected compared to our previous report[30]. Such enlarged voltage hysteresis indicates a sluggish redox reaction kinetics brought about by increasing the sulfur loading. In contrast, the plateau at 2.1 V in the discharge profile of the MTG/S cathode in Fig. 3b becomes very short and imperceptible, indicating a limited NaPS formation and dissolution in the MTG/S cathode. Also, the MTG/S cathode could exhibit a low voltage hysteresis of 0.73 V. These features suggest a promoted kinetics of sulfur redox reaction in MTG/S cathode. Interestingly, a new plateau at 0.9 V was observed in the discharge profile attributed to the sodiation of $MoTe_2$ to $Na_xMoTe_2$. In addition, a new plateau at 1.05 V was detected in the charging process corresponding to the desodiation of $Na_xMoTe_2$ to $MoTe_2$. These behaviors match well with the voltage profile of $MoTe_2$ in Fig. S2 and previous reports[39]. The sodiation/desodiation behaviors of MTG significantly promote the kinetics of sulfur redox reaction.

To further identify the intercalation-conversion electrochemistry in MTG/S cathode, cyclic voltammetry (CV) characteristics were

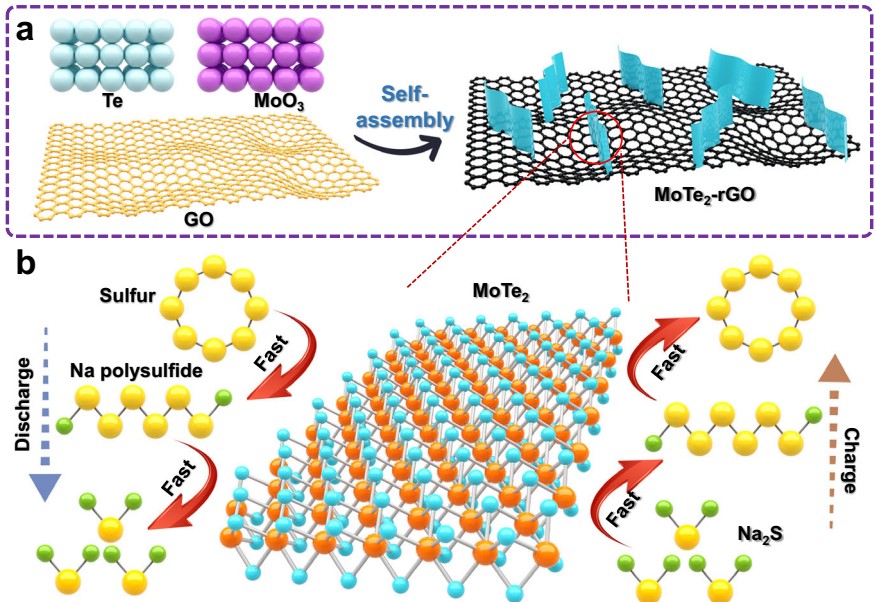

**Fig. 1 | Host design strategy. a** Illustration of the synthetic route of MTG. **b** Fast electrochemical reaction kinetics of Na-S batteries with MTG.

obtained in the voltage range between 0.8 and 2.8 V. The CVs were recorded at a scan rate of 0.1 mV s$^{-1}$, as shown in Fig. 3c, d. The CV curve of C/S cathode shows two representative cathodic peaks at 2.01 and 1.68 V that are attributed to the reduction of sulfur to NaPS and Na$_2$S, respectively. The anodic peaks at 2.08 and 2.30 V are owing to, respectively, the coupled conversion from Na$_2$S to NaPSs and sulfur. As shown in Fig. 3d, the two cathodic peaks related to the sulfur conversion also can be detected. Compared to those of the C/S cathode, the peaks of the MTG/S cathode show a positive shift and become more distinguished, displaying smaller current response and larger polarization. The above results demonstrate that the MTG/S cathode enables a faster redox reaction process and low intrinsic resistance[40]. Additionally, a new cathodic peak at 0.89 V and an anodic peak at 1.06 V appear in the CV curve of MTG/S cathode, which are related to, respectively, the sodiation of MoTe$_2$ to Na$_x$MoTe$_2$ and desodiation of Na$_x$MoTe$_2$ to MoTe$_2$. These electrochemical behaviors match well with the voltage profile in Fig. 3b.

Static adsorption of polysulfides can effectively reveal the affinity of host toward sulfur species, forming during the sulfur conversion in Na-S batteries. As shown in Fig. 3e, the ultraviolet-visible (UV–vis) spectrum of Na$_2$S$_6$ solution exhibits three broad peaks at 239, 283, and 421 nm, corresponding to the S$_6^{2-}$, S$_6^{2-}$, and S$_4^{2-}$ species, respectively. After contacting with carbon for 24 h, the UV–vis spectrum of the solution shows only a slight change. As a comparison, the characteristic peaks of polysulfides in the spectrum of the solution contacting with MTG completely vanish, indicating a strong adsorption ability of MTG. Visual adsorption experiments were carried out in Na$_2$S$_6$ solution to evaluate the chemical anchoring ability of MTG toward polysulfides. As shown in Fig. S3, the polysulfide solution containing MTG is completely transparent after adsorption. In contrast, the polysulfide solution containing C remains light-yellow after adsorption, indicating the much stronger absorptivity of MTG toward polysulfides than that of C. When pure MoTe$_2$ is used as a sulfur host, it can also be used as a polysulfide absorber (Fig. S3). However, the electronic conductivity of MoTe$_2$ is not as high as graphene. Therefore, using pure MoTe$_2$ may not be as beneficial to improve the sulfur redox kinetics. The strong coupling of graphene sheets with MoTe$_2$ nanosheets enhances ionic and electronic transport simultaneously, which is important to achieve a fast sulfur redox. In addition to the adsorption capability of polysulfide, sulfur catalysis also plays an important role in the enhancement of sulfur redox. The liquid-solid nucleation behaviors of sodium

sulfide (Na$_2$S) on the host can reflect the kinetics of sulfur redox reaction, involving the liquid-solid phase conversion of sulfur species. Here, comparative chronoamperometry curves are illustrated in Fig. 3f. The cell with MTG delivers a higher current response of 0.4 mA and a larger deposition capacity of 161.3 mA h g$_s^{-1}$ than those in the cell with carbon (0.2 mA and 43.7 mA h g$_s^{-1}$, respectively). Such results suggest the superior capability of MTG for promoting the conversion from polysulfide into solid Na$_2$S.

## Electrochemical performance

To demonstrate the effectiveness of the intercalation-conversion hybrid cathode, the C/S or MTG/S cathode was paired with Na anode in LHCE and assembled for Na-S cells. Figure 4a shows the voltage profiles of Na‖MTG/S cell at various cycles at 0.1 C rate (1 C = 1675 mA g$_s^{-1}$). In the initial discharge cycle, a short slope at 2.0 V related to the NaPS formation is detected from the voltage profile in Fig. 4a. In addition to the NaPS formation slope, a unique plateau of 0.9 V is observed, corresponding to the sodiation process of MoTe$_2$ to Na$_x$MoTe$_2$. In the charge curve, the plateau at 1.0 V associated with desodiation of Na$_x$MoTe$_2$ to MoTe$_2$ can also be detected. All these electrochemical behaviors matched well with the early discussions. After the initial cycle, the NaPS formation slope becomes negligible, indicating a significant suppression of NaPS formation and shuttling in MTG/S cathode. Notably, the plateaus related to the sodiation of MoTe$_2$ to Na$_x$MoTe$_2$ and desodiation of Na$_x$MoTe$_2$ to MoTe$_2$ still can be observed even after 350 cycles, suggesting the MTG has a good activity after an extensive sodiation/desodiation process. The profiles remain stable after the initial cycle, indicating this sulfur reaction pathway in MTG/S is highly reversible under the conditions studied.

The Na-S cell with MTG/S cathode delivers an initial discharge capacity of 1081 mA h g$_s^{-1}$ as shown in Fig. 4b. Even after 350 cycles, the capacity remains at 901 mA h g$_s^{-1}$, corresponding to a capacity fade of as low as 0.05% per cycle. The Coulombic efficiency at 350 cycles is 100%, which indicates little NaPS shuttling in the MTG/S cathode. It should be noted that the sulfur loading is 3.5 mg cm$^{-2}$, and the electrolyte-to-sulfur (*E/S*) ratio is controlled to be 7 µl mg$^{-1}$. The cell with C/S cathode shows poor cycling performance with a rapid fading within 100 cycles, and finally exhibits a capacity of 456 mA h g$_s^{-1}$ after 246 cycles. The capacity based on the total mass of electrode materials is calculated and presented in Fig. S4, which shows the cycling performance improvement of the electrode. Such significant

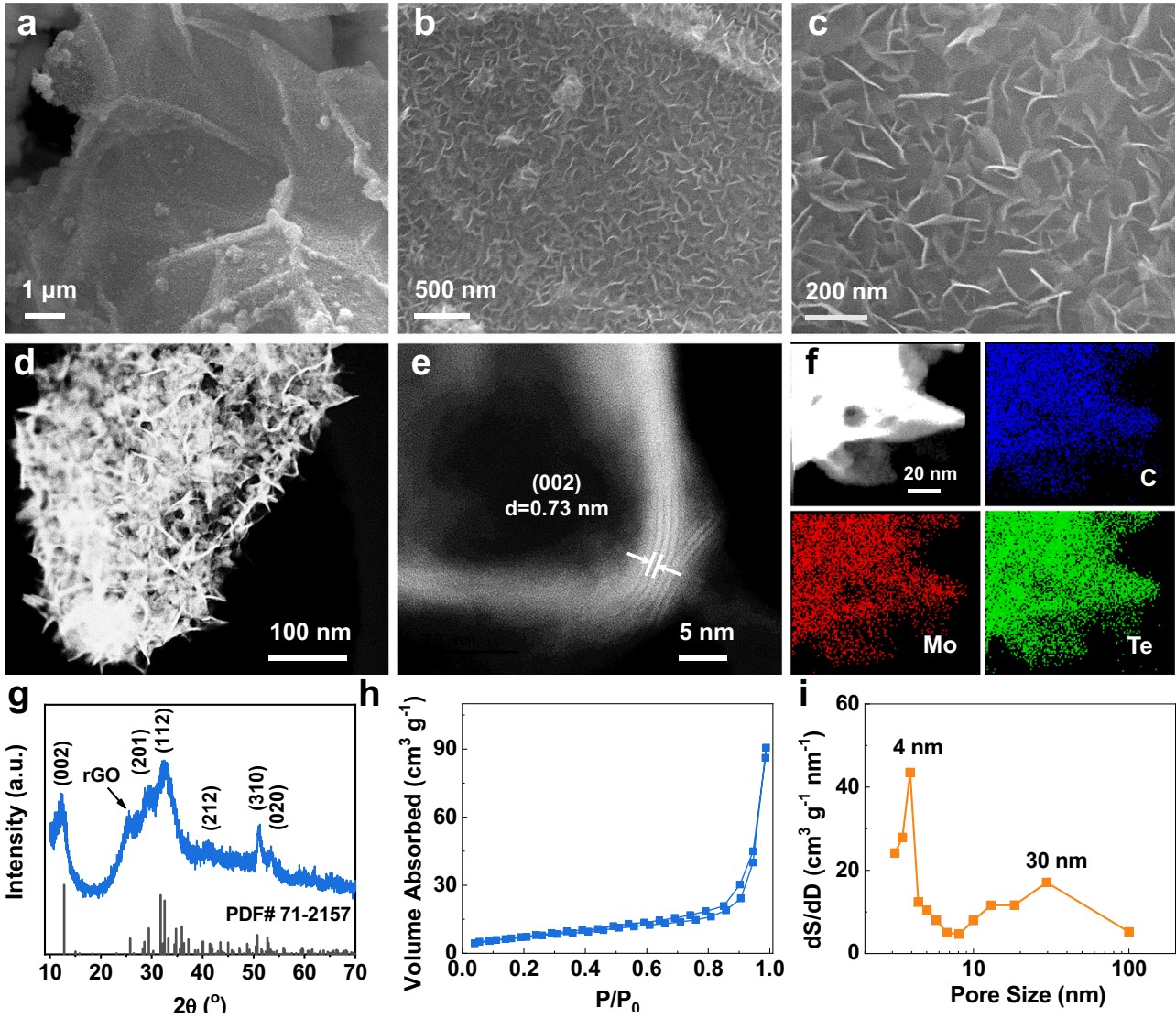

**Fig. 2 | Characterization of MTG. a–c** Scanning electron microscopy images of MTG. **d** TEM image of MTG. **e** High-resolution TEM images of MTG. **f** The TEM image of MTG and its corresponding elemental mapping. **g** XRD pattern of MTG. **h** N$_2$ adsorption/desorption isotherm and (**i**) pore size distribution of MTG composite.

improvement in the specific capacity and lifespan in MTG/S cathode compared to these of C/S cathode further evidence the advantages of MTG in suppressing the NaPS formation and shuttling.

To further identify the role of the sodiation/desodiation of MoTe$_2$ behavior during cycling, the MTG/S cathode was tested in a limiting potential window between 0.8 and 2.8 V. With this potential limit, the continual sodiation/desodiation of MoTe$_2$ was excluded. As shown in Figs. S5 and S6, the capacity of the MTG/S cathode reduces to 637 mA h g$_s^{-1}$ after 100 cycles. The reduction of the capacity can be attributed to two factors. First, without the sodiation/desodiation of MoTe$_2$, the kinetic of the redox was limited, leading to lower utilization of the active material. Second, the sulfur conversion becomes incomplete, resulting in a lower MTG capacity. These results reconfirm the advantage of the intercalation-conversion hybrid cathode.

The MTG/S cathode was further tested at a high current density of 1 C to reveal the kinetics of the reaction and structural stability of the cathode. Figure 4c shows that the MTG/S can still deliver a discharge capacity of 637 mA h g$_s^{-1}$ even after 900 cycles, corresponding to a capacity decay rate of 0.02% per cycle. MTG/S cathode also exhibits improved rate performance at various rates from 0.1 C to 2 C. As shown in Fig. 4d, MTG/S cathode delivers a high specific capacity of 1039, 951,

897, 843, 819, 806, and 694 mA h g$_s^{-1}$, respectively, at 0.1C, 0.2C, 0.3C, 0.5C, 0.8C, 1C, and 2C, suggesting fast reaction kinetics in MTG/S cathode. Furthermore, when the current density switches back to 0.1C, the specific capacity can recover to 1007 mA h g$_s^{-1}$, indicating a highly reversible reaction in the MTG/S cathode.

Higher sulfur loading and higher areal capacities are essential for high energy density. Given this, cells with MTG/S cathode were further tested at higher sulfur loadings to support higher energy density. A further cycling evaluation on MTG/S cathode was performed at 0.1C with higher sulfur loading of 4.7 and 5.8 mg cm$^{-2}$. As shown in Fig. 4e, with a sulfur loading of 4.7 mg cm$^{-2}$, MTG/S cathode delivers an initial discharge capacity of 1000 mA h g$_s^{-1}$ and maintains good stability over 100 cycles. Furthermore, MTG/S cathode with a high sulfur loading of 5.8 mg cm$^{-2}$ can still deliver an initial high capacity of 887 mA h g$_s^{-1}$, equivalent to an areal capacity of 5.1 mA h cm$^{-2}$. Even after 100 cycles, the MTG/S cathode exhibits good cycling durability with capacity retention of up to 73.8%. The ability of MTG/S cathode to operate under these conditions highlights the abilities of MTG compared to the state-of-the-art hosts, as summarized in Table S1. Such upbeat characteristics point to the alluring host architecture of MTG for high-performance Na-S batteries.

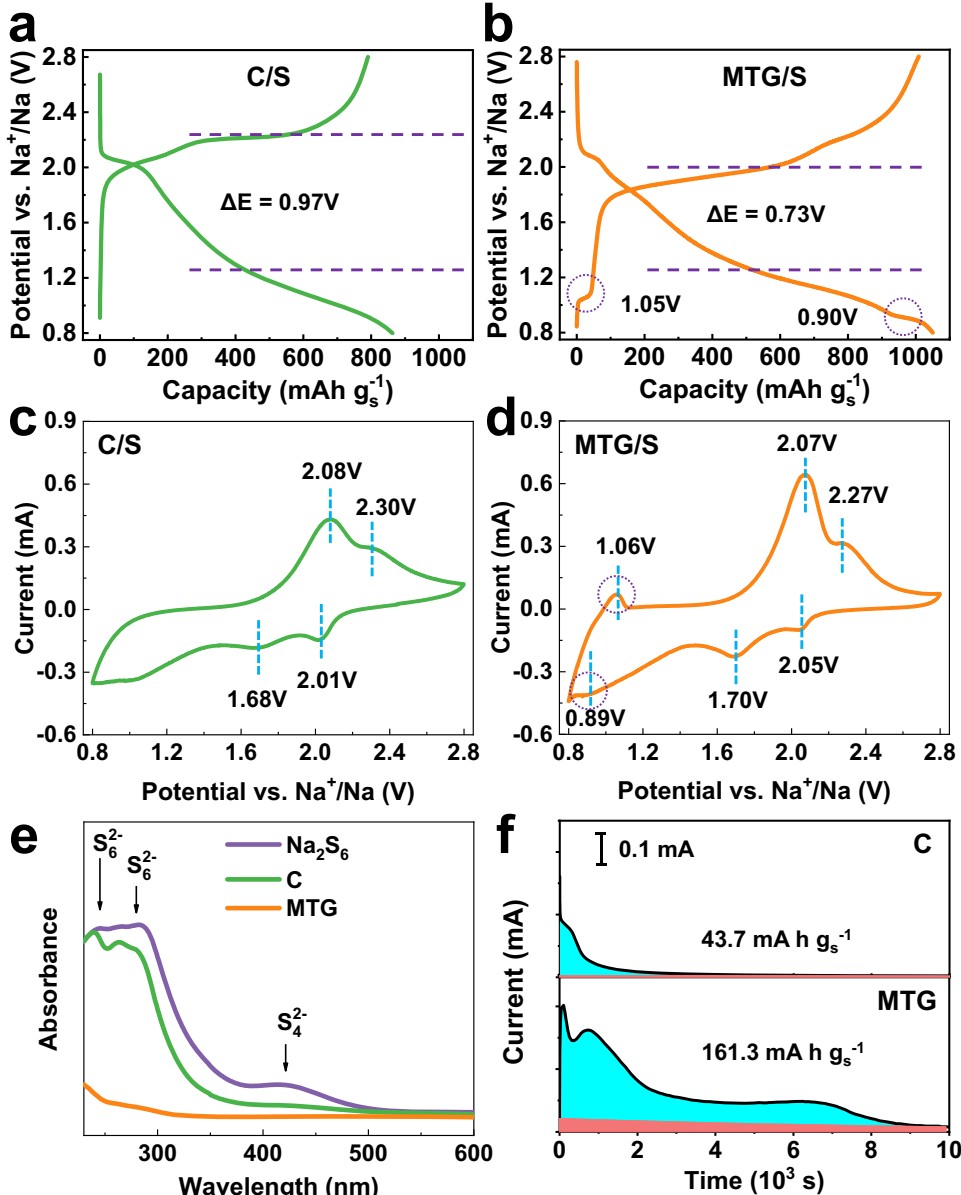

**Fig. 3 | Investigation of the interaction between MTG and NaPS.** The voltage profile of (**a**) C/S cathode and (**b**) MTG/S cathode. The CV curves of (**c**) C/S cathode and (**d**) MTG/S cathode. **e** UV–visible absorption spectra of the $Na_2S_6$ solution after adsorbing by C or MTG. **f** Chronoamperometry curves of $Na|Na_2S_8$ cells, showing the kinetics of liquid-solid $Na_2S$ deposition.

## Delineation of the sulfur transformation and interfacial chemistry

Synchrotron-based operando X-ray diffraction (XRD) was conducted to experimentally assess the sulfur transformation in different hosts. As shown in Fig. S7, a window coin cell was designed for the operando testing, which has been applied in our previous reports[41,42]. The details of the assembled cells are described in the "Methods" section. Synchrotron X-rays went through all components within a window coin cell, and their diffraction patterns were combined into a single XRD pattern. Figure 5 shows an operando XRD result of sulfur cathodes for Na-S window coin cells. The cells were tested at 0.1C in the voltage range of 0.8 and 2.8 V for one full discharge-charge cycle, during which the XRD signal was simultaneously recorded as reflected by the contour plot next to the discharge-charge profile. In the contour plot of the cell with C/S cathode (Fig. 5a), a set of dominant peaks of α-$S_8$ (JCPDS No. 008-0247) can be clearly detected at the beginning of discharge and remains detectable at high intensity during the entire

discharge process, indicating the incomplete conversion of sulfur to $Na_2S$. This is owing to the sluggish kinetics of sulfur redox reaction in C/S cathode. Such incomplete conversion of sulfur well explains the low utilization of active material and low capacity of C/S cathode. In addition to the sulfur peak, one dominant peak at 1.66° corresponding to the (031) of $Na_2S_5$ (JCPDS No.044-0823), which can be a representative of the NaPSs, can be detected during the whole discharge process, demonstrating the inability of carbon to promote NaPS conversion and resulting the severe NaPS shuttling in the cell with C/S cathode. In sharp contrast, in the case of the MTG/S cathode (Fig. 5b), the strong α-$S_8$ peak was detected in the contour plot at the beginning of the discharge and gradually vanished along with the discharge process. This suggests the efficient transformation of sulfur to $Na_2S$. It is noted that no obvious peak associated with MTG can be identified because of its much lower intensity compared to sulfur. Moreover, no pronounced $Na_2S_5$ peak is observed during the entire discharge process, indicating the effective suppression of NaPS formation and

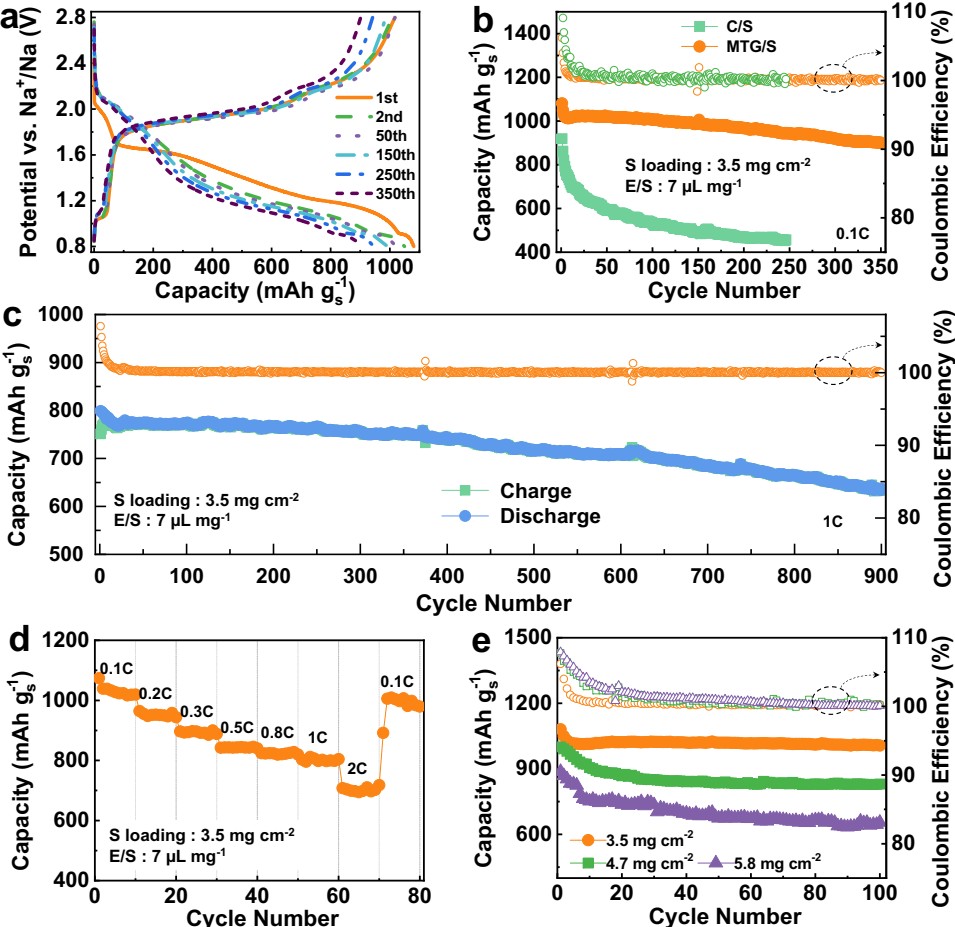

**Fig. 4 | Electrochemical behavior of various cathodes. a** Voltage profiles of MTG/S cathode at 0.1C rate with a high sulfur loading of 3.5 mg cm⁻² in the voltage range between 0.8 and 2.8 V. **b** Cycling performance of MTG/S cathode and C/S cathode at 0.1C rate. **c** Cycling performance of MTG/S cathode at 1C rate. **d** Rate performance of MTG/S cathode at various C rates. **e** Cycling performance of MTG/S cathode at 0.1C rate with various sulfur loading.

shuttling in MTG/S cathode. During charge, the β-S₈ phase gradually appears, which indicates that, by this point, the polar MTG facilitates efficient electron transfer and promotes redox to sulfur, assisting in its nucleation. This results in solid and clear peaks for β-S₈ (JCPDS No. 071-0137) by the end of the charge. Thus, this investigation provides clear insights into the efficient polysulfide mediation provided by MTG.

To further investigate the interfacial chemistry of the cathode, time-of-flight secondary ion mass spectrometry (ToF-SIMS) was performed, which can easily verify the spatial distribution of sulfur in hosts. Figure 5c shows the ToF-SIMS depth profiles of S⁻ secondary ion in various cycled S cathodes at 0.1C for 100 cycles. The depth profile of cycled C/S cathode exhibits a high intensity in the initial sputtering time of 200 s, then rapidly fades to very low intensity, and finally remains stable at 20% of its peak intensity. This behavior indicates a sulfur migration layer owing to the NaPS shuttling. In sharp contrast, the intensity of the MTG/S cathode quickly reaches its peak and maintains a high value after that. Such a homogeneous distribution of sulfur suggests the excellent confinement of sulfur species within MTG.

The thickness of the sulfur migration layer in C/S cathode can further be revealed by the 3D visualization of the architecture evolution in Fig. 5d. Due to the weak immobilization of carbon toward NaPSs, most sulfur species migrate outside the framework of the carbon host, and only a small amount of sulfur is residual in the carbon host. Such migration of the sulfur results in huge active material loss and consequently leads to the poor lifespan of the Na-S batteries. In

sharp contrast, the MTG/S maintains a uniform sulfur distribution over 100 cycles. No sulfur migration was observed in the 3D visualization of the architecture evolution of cycled MTG/S cathode in Fig. 5e. Such uniform sulfur distribution in cycled MTG/S cathode further suggests the MTG architecture remarkably immobilized sulfur species within the cathode structure through physical and chemical molecular interactions. As shown in Fig. S8, the MT/S shows a uniform sulfur distribution after 100 cycles, which is similar to that of MTG/S. The comparative cross-sectional morphologies of cycled Na in Na–S cells with MTG and C/S cathodes are shown in Fig. S9 to demonstrate the advantage of MTG further. As shown in Fig. S9a, the surface of cycled Na paired with MTG/S cathode is smooth without cracking. In contrast, the surface of the Na paired with C/S cathode is covered by a thick uneven layer with many cracks (Fig. S9b), which may be attributed to the NaPS shuttling in the C/S cathode.

In summary, we develop an intercalation-conversion hybrid cathode that enables a shuttle-less sulfur cathode and improves sulfur redox kinetics in ambient-temperature Na-S batteries. The intercalation-type MTG not only serves as multiple adsorptive sites to chemically confine sulfur species and catalyze the sulfur conversions toward fast and durable sulfur electrochemistry, but also offers good short-range ion conduction for sulfur redox reactions by sodiation/desodiation process of MTG. Such an intercalation-conversion pathway further inhibits polysulfide dissolution and shuttling and greatly suppresses parasitic reactions with the Na anode. These simultaneous benefits allow for a long lifespan of 900 cycles with a capacity decay

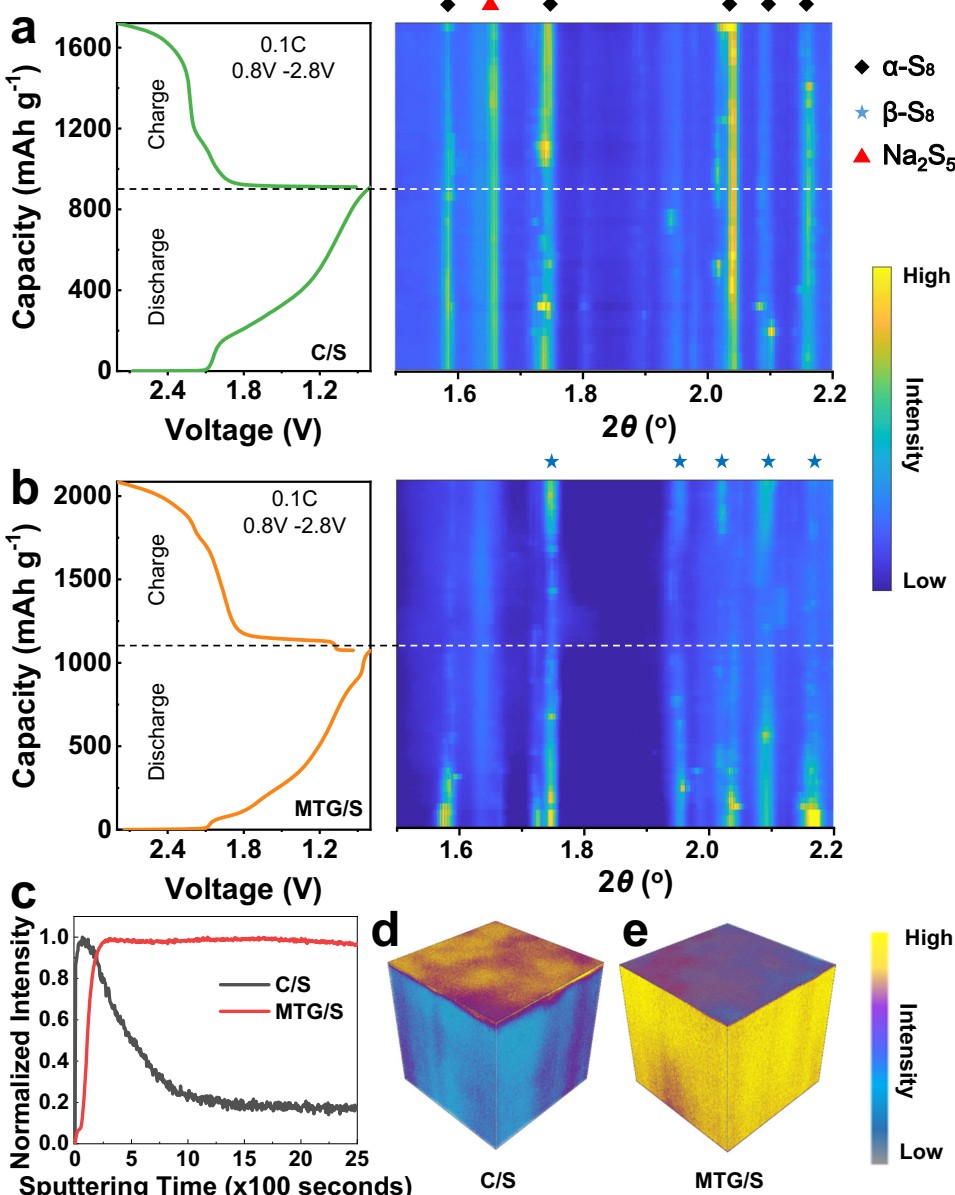

**Fig. 5 | Advanced characterization of Na-S batteries with different cathodes.** A charge and discharge curve of a Na-S coin cell and the corresponding diffraction patterns with (**a**) C/S and (**b**) MTG/S cathodes. Diamond, triangle, and star symbols represent α-S$_8$, Na$_2$S$_5$, and β-S$_8$. (**c**) ToF-SIMS depth profiles of S$^-$ secondary distribution of cycled electrodes after 100 cycles at 0.1C. The 3D rendering of S$^-$ secondary distribution of cycled (**d**) C/S and (**e**) MTG/S after 100 cycles.

rate of 0.02% per cycle under a high sulfur loading of 3.5 mg cm$^{-2}$ and a lean electrolyte condition ($E/S$ = 7 μl mg$^{-1}$).

## Methods

### Preparation of the MTG
Graphene oxide (GO) was prepared by a modified Hummers' method similar to that presented in the previous report[43]. A 5 g amount of graphite and 20 g of KMnO$_4$ were added into 200 ml of concentrated H$_2$SO$_4$ and stirred for 2 h at 35 °C. Then 400 ml of deionized water (DI water) was slowly added into the mixed solution, and the temperature of the solution was kept at 95 °C. Next, 5 ml of H$_2$O$_2$ was added into the previous solution when its temperature decreased to 60 °C. Afterward, 100 ml of diluted HCl was added. Finally, the GO solution was filtrated several times with DI water and then dried. A one-pot solvothermal strategy synthesized MTG. 0.2 g of GO was well dispersed in 45 ml N, N-dimethylformamide (DMF) by ultrasonication probe (1000 W) for 30 min to form a uniform suspension. 0.765 g of Te powder, 0.432 g of

MoO$_3$, and 15 ml of hydrazine hydrate (35%) were added above the suspension. The mixture was stirred for 24 h, transferred into a 100 ml autoclave, and then heated at 220 °C for 24 h. After cooling down naturally, the formed MTG was vacuum filtered and washed several times with deionized water and ethanol before being dried by freeze-drying.

### Preparation of the sulfur cathode
The sulfur cathode in this work was obtained by a modified melt-diffusion method[35]. The MTG or Ketjen Black carbon was mixed with an appropriate amount of sulfur with a mass ratio of 1:9, and the mixture was heated at 155 °C for 12 h in a sealed vial under an Ar atmosphere. Then, the as-obtained product was heated at 200 °C for 30 min in a flowing Ar atmosphere in a tube furnace to remove the redundant sulfur outside of the MTG or Ketjen Black carbon. The sulfur content in the composite was calculated to be ~87 wt%, which was determined by comparing the weight of the composites before and after melt-diffusion method.

## Preparation of the electrolyte

Sodium bis(fluorosulfonyl)imide (NaFSI, Solvionic Corporation) was dried in a vacuum oven for 24 h at 60 °C. 1,2-Dimethoxyethane (DME, Sigma-Aldrich) was dried with molecule sieves for 3 days. The electrolyte was prepared inside an Ar-filled glove box. Particularly, DME was mixed with NaFSI at a molar ratio of 1:1.2 and stirred over 3 h to obtain transparent solutions. Then, 1,1,2,2-tetrafluoroethyl 2,2,3,3-tetrafluoropropyl ether (TTE) was added to the above solutions with the same molar amount as NaFSI.

## Characterizations

The morphology investigation was performed with a scanning electron microscope (FEI Quanta 650 SEM operated at 20 kV) and an energy-dispersive X-ray (EDX) spectrometer to detect the elemental signals. TEM images were collected with a TEM field emission from JEOL 2010F powered at 200 kV. The adsorption of the host toward the polysulfide ($Na_2S_6$ was chosen as a representative) was measured with a UV–vis spectrometer (Cary 5000) with baseline correction (Varian).

## Synchrotron X-ray diffraction (XRD)

Synchrotron operando XRD measurements were collected at the 11-ID-C beamline at the Advanced Photon Source (APS) at Argonne National Laboratory. The X-ray energy was 105.7 keV, and the beam spot size was 0.5 mm by 0.5 mm. A Perkin Elmer area detector was used to collect the 2D diffraction images, the sample-to-detector distance was calibrated using a $CeO_2$ standard, and the images were integrated into 1D diffraction patterns using the GSAS-II package[44]. XRD data were collected continuously in transmission mode by rastering back and forth between cells with a collection time of 2 min per pattern, during which the cells were cycled at 0.1C rate from 0.8–2.8 V for one full charge-discharge cycle. House-built CR2016-type window coin cells were assembled inside a glovebox with $H_2O$ and $O_2$ levels lower than 1 ppm. The schematic of the window coin cell is shown in Fig. S7. Both the top casing and the bottom casing were punched with a 3 mm in diameter window to allow the penetration of X-ray. The windows were sealed with Al foil (12 μm thick) and Cu foil (9 μm thick), respectively, on the cathode side and the anode side. An epoxy composed of Eccobond 45 and Catalyst 15 with a ratio of 1:1 in volume was applied to allow a reliable sealing, as demonstrated in previous publications[41,45]. Two discs of Na metal were used as both the ion source and as the spacer to maintain good mechanical contact with the sulfur electrode. These house-built cells were assembled in our lab and shipped to APS for operando characterization.

## Electrochemical measurements

The electrodes for the liquid-solid reaction kinetics study contained 60 wt% MTG or Ketjen Black carbon, 20 wt% Super-P, and 20 wt% sodium carboxymethyl cellulose (CMC) binder. The main reason CMC is used as the binder for the MTG/S electrode is that the Na-ions present in CMC could potentially provide some ionic conductivity, greatly improving the reversible capacity of Na-S cells. In addition, compared with PVDF, the electrode with CMC as a binder shows enhanced electrochemical performance, consistent with previous reports[46]. The sodium foil was prepared manually during cell assembly. The thickness of sodium foils is around 600 μm. The Na||Cu cells were constructed using Cu foil (99.95% purity, 9 μm thickness, Guangdong Canrd New Energy Technology Co., Ltd.) as working electrode, Na metal as counter electrode. Following a previously validated method, the Na||$Na_2S_6$ cells were constructed for liquid-solid reaction kinetics study with chronoamperometry. In detail, the working electrode (MTG or Ketjen Black carbon) was assembled with Celgard 2500 separator and a Na foil anode inside an Ar-filled glove box. A 30 μl of 0.2 M $Na_2S_6$/1 M NaTFSI-tetraglyme catholyte and 20 μl of 1 M NaTFSI-tetraglyme anolyte were used. Cells were discharged galvanostatically to 1.3 V, followed by a potentiostatic discharging at 1.25 V for about 10,000 s. For

the standard Na-S batteries, each electrode contained 80 wt% sulfur composite, 5 wt% Super-P, 5 wt% multi-walled carbon nanotubes, and 10 wt% sodium carboxymethyl cellulose (CMC) binder. All electrodes were cut into small plates with a diameter of 10 mm. The sulfur loading in each electrode was ~70 wt%. Coin-type (CR2032) cells were assembled in an Ar-filled glove box with sodium metal as the anode. Celgard 2500 was used as the separator. The electrolytes were prepared as in the abovementioned procedures. The *E/S* ratio in the coin cells was 7 μl mg$^{-1}$. An Arbin battery cycler was used to conduct the cycling performance between 0.8 and 2.8 V. Cyclic voltammetry (CV) measurements were evaluated with a VoltaLab PGZ 402 Potentiostat with various scan rates in the potential range of 0.8 and 2.8 V. All the cells were tested at room temperature. In this work, the specific capacity values were calculated based on sulfur mass ($g_s$).

## Data availability

The datasets generated during and/or analyzed during the current study are available from the corresponding author upon request.

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

## Acknowledgements

This work was supported by the U.S. Department of Energy, Office of Basic Energy Sciences, Division of Materials Science and Engineering under award number DE-SC0005397.

## Author contributions

J.H. designed, performed the experiments, and wrote the manuscript. A.B. performed the high-loading coin cell experiments. H.C. and L.S. performed the Synchrotron operando XRD and corresponding analysis. A.M. supervised the project and edited the manuscript. All authors discussed the results and reviewed the manuscript.

## Competing interests

The authors declare no competing interests.
