## [Peer Review File · Nature Communications]

nature portfolio

Peer Review FileREVIEWER COMMENTS

Reviewer #1 (Remarks to the Author):

He et al reported an intercalation-conversion hybrid strategy for high-performance sodium-sulfur batteries where MoTe₂ nanosheets-graphene composites were employed as an effective sulfur host with fast sodium-ion intercalation-deintercalation and catalytic activity. Impressively, the hybrid cathode exhibits a high capacity and excellent cycling performance under high sulfur loading and lean electrolyte condition, which are quite challenging for Na-S batteries. Such practical performance could be a benchmark for recent RT Na-S fields. I have some comments.

1. From the authors' viewpoint, what applications should RT Na-S batteries be applied for in the future? What are the advantages of Na-S batteries over Li-S batteries besides the cost?
2. Some important references for hybrid strategies in Li-S batteries are missing, like ACS Energy Lett. 2018, 3, 3, 568–573, Nat Energy 4, 374–382 (2019). It is difficult for me to find the electrolyte used in this work based on the only information "were tested in the localized high-concentration electrolyte (LHCE) as reported in our previous work". Please add related references.
3. Is there any visual evidence for the suppressed shuttling effect besides Fig. 3e?
4. Why the authors choose MoTe₂, not MoS₂ which is cheaper?
5. In Supplementary Table 1, the authors compared recent Na-S literature. I recommend using sulfur content in the cathode, not in composite to reflect the actual active material contained in the cathode.
6. Could the authors characterize the Na-metal foils cycled in the Na-S cells with different cathodes? It is quite helpful for further demonstrating the suppressed shuttling effect.
7. The compatibility of the electrolyte used in this work should also be further evaluated by Na-Cu or Al cells without the effect of sulfur cathodes.

Reviewer #2 (Remarks to the Author):

The authors in their manuscript "Intercalation-type catalyst for high-performance sodium-sulfur batteries" have shown advanced intercalation-conversion hybrid cathode by coupling the intercalation-type catalyst of MoTe₂ with conversion-type active material of sulfur to achieve a high-performance Na-S batteries. Similar work has been already done for Li-S battery; however, use of MoTe₂-graphene sheet for Na-S is interesting. The fundamental understanding of the electrocatalysis of MoTe₂ is poorly described in this work. Besides, there are several questions raise:

1. What about pure MoTe₂ as sulfur host? Will it work as the polysulphide absorber such nicely without graphene too?
 2. What is the contribution of graphene and MoTe₂ separately in Na-S for cyclic stability improvement?
 3. Also the synchrotron X-ray diffraction suggests uniform sulfur distribution for MTG after 100 cycles. What about pure MT? will it be same as MTG?
 4. Usually cathodes are prepared with pvdf binder in case of metal-sulfur batteries. Is there any particular reason to use CMC as binder for MoTe₂ instead of pvdf? What will be the performance with pvdf compared to cmc?
- These questions will give insight into this work.

RESPONSE TO REVIEWERS' COMMENTS

REVIEWER 1

General Comment: *He et al reported an intercalation-conversion hybrid strategy for high-performance sodium-sulfur batteries where MoTe₂ nanosheets-graphene composites were employed as an effective sulfur host with fast sodium-ion intercalation-deintercalation and catalytic activity. Impressively, the hybrid cathode exhibits a high capacity and excellent cycling performance under high sulfur loading and lean electrolyte condition, which are quite challenging for Na-S batteries. Such practical performance could be a benchmark for recent RT Na-S fields. I have some comments.*

Response to General Comment: We appreciate the reviewer very much for the positive feedback and valuable comments/suggestions, which were very useful for us to improve our manuscript.

Comment 1: *From the authors' viewpoint, what applications should RT Na-S batteries be applied for in the future? What are the advantages of Na-S batteries over Li-S batteries besides the cost?*

Response to Comment 1: We thank the reviewer for this comment. In our viewpoint, RT-Na/S batteries would be promising candidates for grid-scale energy-storage in the conceivable future. In addition, Na-S batteries with rich resources of both sodium and sulfur have also been considered as a supplement to Li-S batteries. In contrast to the limited resources (0.0018 wt% in the earth's crust) of Li, Na is more abundant (> 2.5 wt% in the earth's crust) [InfoMat, 2022, 4(5): e12291].

Comment 2: *Some important references for hybrid strategies in Li-S batteries are missing, like ACS Energy Lett. 2018, 3, 3, 568–573, Nat Energy 4, 374–382 (2019). It is difficult for me to find the electrolyte used in this work based on the only information “were tested in the localized high-concentration electrolyte (LHCE) as reported in our previous work”. Please add related references.*

Response to Comment 2: We thank the reviewer for the suggestions. The relevant work has now been added as new references 13 and 14 on page 2 in the revised manuscript. The LHCE work is cited as reference 22 on page 3 in the revised manuscript.

13. Chung, S., Luo, L., Manthiram A. TiS₂-polysulfide hybrid cathode with high sulfur loading and low electrolyte consumption for lithium-sulfur batteries. *ACS Energy Lett.* **3**: 568-573 (2018).

14. Xue, W. et al. Intercalation-conversion hybrid cathodes enabling Li-S full-cell architectures with jointly superior gravimetric and volumetric energy densities. *Nature Energy.* **4**: 374-382 (2019).

22. He, J., Bhargav, A., Shin, W. & Manthiram, A. Stable Dendrite-Free Sodium–Sulfur Batteries Enabled by a Localized High-Concentration Electrolyte. *J. Am. Chem. Soc.* **143**: 20241-20248 (2021).

In LHCE, the sulfur redox process is found to change from the conventional dissolution-precipitation chemistry to a quasi-solid-state reaction. The quasi-solid-state reaction suppresses NaPS shuttling and ensures fast reaction kinetics. Meanwhile, LHCE promotes the formation of a stable SEI on Na metal, effectively prohibiting dendritic Na growth. However, there is still a challenge of sluggish kinetics when using LHCE. In this work, by introducing the intercalation-type catalyst, such issue has been overcome.

Comment 3: Is there any visual evidence for the suppressed shuttling effect besides Fig. 3e?

Response to Comment 3: We thank the reviewer for this comment. Visual adsorption experiments were carried out in Na_2S_6 solution to evaluate the chemical anchoring ability of MTG toward polysulfides. As shown in Figure R3 below, the polysulfide solution containing MTG is completely transparent after adsorption. In contrast, the polysulfide solutions containing C remain light-yellow after adsorption. This indicates the much stronger absorptivity of MTG toward polysulfides than that of C. When pure MoTe_2 (MT) is used as a sulfur host, it can also be used as a polysulfide absorber. However, the electronic conductivity of MoTe_2 is not as high as graphene. Therefore, using pure MoTe_2 may not be as beneficial to improve the sulfur redox kinetics. The strong coupling of graphene sheets with MoTe_2 nanosheets enhances both ionic and electronic transport simultaneously, which is important to achieve fast sulfur redox. Figure R3 below has been added as Figure S3 in the revised supporting information. We have included the corresponding discussion on page 9 in the revised manuscript.

Figure R3. Photograph of the sealed vials of a Na_2S_6 /DME solution after contacting with C, pure MoTe_2 (MT), and MTG.

Comment 4: Why the authors choose MoTe_2 , not MoS_2 which is cheaper?

Response to Comment 4: We thank the reviewer for this comment. As one of the many transition-metal dichalcogenides (TMDs), MoTe_2 possesses a larger van der Waals interlayer distance (0.392nm) than MoS_2 (0.347nm). The large interlayer space can easily accommodate the large Na-ions, resulting in a better TMD structural stability during the cycling process. Moreover, the electronic conductivity of MoTe_2 is 1.8 S cm^{-1} , which is nine times higher than that of MoSe_2 (0.2 S cm^{-1}) [phys. stat. sol. (b), 79: 713]. Such results clearly demonstrate that MoTe_2 can further promote the transport of electrons. We have included the corresponding discussion on page 3 in the revised manuscript.

Comment 5: In Supplementary Table 1, the authors compared recent Na-S literature. I recommend using sulfur content in the cathode, not in composite to reflect the actual active material contained in the cathode.

Response to Comment 5: We thank the reviewer for the suggestion, which helps to improve the quality of our manuscript. The sulfur contents in the cathodes have been revised to reflect the actual active material.

Table R1. A comparative analysis of the Na-C cells with MTG/S and the state-of-the-art Na-S cells reported in the literature

Cathode	Electrolyte	S content/ loading in Electrode (wt.% / mg cm ⁻²)	E/S ratio ($\mu\text{L cm}^{-2}$)	Current density (C-rate)	Reversible capacity (mA h g ⁻¹)	Cycle life	Capacity fading rate (%)	Reference
		69.6 / 3.5	7	0.1C	1081	350	0.05	
	LHCE							
MTG/S	(DME : NaFSI : TTE = 1 : 1.2 : 1)	69.6 / 3.5	7	1C	799	900	0.02	This work
		69.6 / 5.8	7	0.1C	887	100	0.26	
C/S								
(Ketjen Black ca rbon/Sulfur)	LHCE	69.6 / 2	15	0.1C	922	300	0.09	R1
covalent-SC	NaClO ₄ in EC/D EC/FEC	26.6 / 1.4	--	0.8C	1,070	600	0.028	R2
S/C (sodiated ha rd carbon anode)	NaClO ₄ + Na ₂ S/P ₂ S ₅ TEGDME	40.5 / 2.4	25	0.05C	920	1000	0.06	R3
ACC-40S	NaClO ₄ in EC/P C/FEC	32.0 / 1	--	0.1C	1492	400	0.06	R4
CN/Au/S	NaClO ₄ in PC/F EC	39.6 / - -	--	0.06C	701	110	0.58	R5
S/C	NaClO ₄ + Na ₂ S/P ₂ S ₅ TEGDME	47.0 / 2	--	0.05C	780	70	0.88	R6
S@MPCF	NaTFSI in PC/F EC/InI ₃	48.8 / 0.36	50	0.1C	1170	500	0.011	R7
S/C	NaClO ₄ + NaNO ₃ in TEGDME	38.4 / 2.5	7.5	--	1000	20	3	R8
Na ₂ S ₆ - CC@Mn O ₂	NaClO ₄ + NaNO ₃ in TEGDME	-- / 1.7	--	0.12C	938	150	0.16	R9
C/S/BTO	NaClO ₄ in EC/D EC	62 / 1.2	--	0.3C	1101	400	0.11	R10
S/C	NaPF ₆ in TEGD ME	40.0 / - -	--	0.1C	776	20	1.7	R11

S@Ni-NCFs	MCS-Li	36 / 0.7	128	1C	431	270	0.17	R12
core-shell ZCS @S	NaClO ₄ in EC/D EC/FEC	45.6/2	--	0.6C	572	--	--	R13
FeS ₂ @NCMS/S	NaClO ₄ in EC/P C/FEC	45.9/--	--	0.06C	1471	300	0.2	R14
S/MoS ₂ /NCS	--	35/--	--	0.6C	427	2800	0.0055	R15
Co ₁ -ZnS/C@S	NaClO ₄ in EC/D EC/FEC	52/--	--	0.6C	620	--	--	R16

Comment 6: *Could the authors characterize the Na-metal foils cycled in the Na-S cells with different cathodes? It is quite helpful for further demonstrating the suppressed shuttling effect.*

Response to Comment 6: We thank the reviewer for the suggestion, which helps to improve the quality of our manuscript. The comparative cross-sectional morphologies of cycled Na in Na-S cells with MTG and C/S cathodes are shown below in Figure R4 to further demonstrate the advantage of MTG. As shown in Figure R4a below, the surface of cycled Na paired with MTG/S cathode is smooth without cracks. In contrast, the surface of the Na paired with C/S cathode is covered by a thick uneven layer with many cracks (Figure R4b), which may be attributed to the NaPS shuttling in C/S cathode. Figure R4 below has been added as Figure S6 in the revised supporting information. We have included the corresponding discussion on page 14 in the revised manuscript.

Figure R4 | Cross-sectional morphologies of cycled Na in Na-S cells with (a) MTG and (b) C/S cathodes.

Comment 7: *The compatibility of the electrolyte used in this work should also be further evaluated by Na-Cu or Al cells without the effect of sulfur cathodes.*

Response to Comment 7: We thank the reviewer for the suggestion, which helps to improve the quality of our manuscript. The LHCE can not only mitigate NaPSs shuttling, but can also protect the Na anode from dendritic growth. As shown in Figure R5 below, the Na||Cu cell with LHCE can deliver a stabilized coulombic efficiency of 95.2% over 50 cycles. Figure R5 below has been added as Figure S1 on page 1 in the revised supporting information. The corresponding discussion has been included on page 7 in the revised manuscript.

Figure R5 | (a) CE of Na plating-stripping using Cu electrodes in LHCE at 1 mA cm^{-2} with an areal capacity of 1 mAh cm^{-2} and (b) the corresponding Na plating/stripping profile.

REVIEWER 2

General Comment: *The authors in their manuscript "Intercalation-type catalyst for high-performance sodium-sulfur batteries" have shown advanced intercalation-conversion hybrid cathode by coupling the intercalation-type catalyst of MoTe_2 with conversion-type active material of sulfur to achieve a high-performance Na-S batteries. Similar work has been already done for Li-S battery; however, use of MoTe_2 -graphene sheet for Na-S is interesting. The fundamental understanding of the electrocatalysis of MoTe_2 is poorly described in this work. Besides, there are several questions raise:*

Response to General Comment: We appreciate the reviewer very much for the positive comments and valuable comments/suggestions, which are very useful for us to improve our manuscript. We feel sorry that we did not state so clearly the novelty of our manuscript, Here, we would like to clearly state the significant novelty and the corresponding description has also been now included on page 2 in the revised manuscript.

To address the challenges of Na-S batteries, intensive research efforts, such as the combination of conductive carbon and sulfur, modification of separators with carbon, and development of sulfide cathodes have been pursued. In these systems, decent electrochemical performance has been illustrated with low sulfur loading ($< 2 \text{ mg cm}^{-2}$). However, recently reported work is mainly based on carbonate electrolytes, where a flooded electrolyte amount is required to fully wet the glass fiber separator. In this regard, it is still difficult to achieve high-performance Na-S batteries under rigorous conditions, including high active material loading and lean electrolyte, which is of crucial importance for practical Na-S batteries. A localized high-concentration electrolyte (LHCE) developed in our previous work enable a stable Na-S battery under a decent amount of electrolyte condition. However, the capacity is only 675 mA h g^{-1} after 300 cycles with a low sulfur loading of 2 mg cm^{-2} at a low rate of 0.1 C, which is far from the theoretical capacity of $1,675 \text{ mA h g}^{-1}$ of Na-S batteries. These results indicate that the utilization of sulfur and the kinetics of the redox need to be further improved.

Herein, for the first time, we introduce a new concept of intercalation-conversion hybrid cathode to dramatically promote the kinetics of sulfur redox and improve the utilization of

sulfur by coupling the intercalation-type catalyst MoTe₂ with the conversion-type active material sulfur in Na-S batteries. In addition, MoTe₂ nanosheets vertically grown on graphene flakes offer abundant active catalytic sites, further boosting catalytic activity for sulfur redox. With the intercalation-conversion strategy and well-designed structure, a high capacity of 1,081 mA h g⁻¹ with a capacity fading rate of only 0.05% per cycle over 350 is realized under high sulfur loading and lean electrolyte condition. In addition, the MTG/S can still deliver a discharge capacity of 637 mA h g⁻¹ even over an impressive 900 cycles, corresponding to a capacity decay rate of 0.02% per cycle, **which is the longest lifespan demonstrated to date**. The fundamental understanding of the electrocatalysis of MoTe₂ is further revealed by *in-situ* synchrotron-based operando energy dispersive X-ray diffraction and *ex-situ* time-of-flight secondary ion mass spectrometry, which provides new insights and opportunities to develop advanced Na-S batteries with highly efficient electrocatalyst for sulfur conversion.

The concept of intercalation-conversion strategy provides a new route for designing novel sulfur cathode for Na-S batteries. Such a concept guides further exploration of promising sulfur cathodes with intercalation-type catalysts. These catalysts will enable high-energy cathodes for developing next-generation Na-S batteries as well as Li-S batteries.

Also, the fundamental understanding of MTG in Na-S batteries and the deep discussion on the electrochemical results collected under practically necessary conditions in this manuscript can provide new insights and opportunities to develop practical Na-S batteries. We believe that our work can motivate the rechargeable battery community to broaden their interest and benefit the energy storage area.

Comment 1: *What about pure MoTe₂ as sulfur host? Will it work as the polysulfide absorber such nicely without graphene too?*

Response to Comment 1: We thank the reviewer for the comment. As shown in Figure R3 below, when pure MoTe₂ is used as a sulfur host, it can also be used as a polysulfide absorber. However, the electronic conductivity of MoTe₂ is not as high as graphene. Therefore, using pure MoTe₂ may not be as beneficial to improve the sulfur redox kinetics. The strong coupling of graphene sheets with MoTe₂ nanosheets enhances both ionic and electronic conductivities simultaneously, which is important to achieve fast sulfur redox. Figure R3 has been added as Figure S3 in the revised supporting information. We have included the corresponding discussion on page 10 in the revised manuscript.

Figure R3 | Photograph of the sealed vials of a Na₂S₆/DME solution after contacting with C, pure MoTe₂ (MT), and MTG.

Comment 2: What is the contribution of graphene and MoTe₂ separately in Na-S for cyclic stability improvement?

Response to Comment 2: We thank the reviewer for the comment. The use of graphene provides conductive support to enhance the conductivity of sulfur-based cathode. The highly conductive graphene skeleton guarantees fast charge transfer, further accelerating the sulfur species conversion reaction. The intercalation-type MoTe₂ can provide fast Na-ion transport ability, which ensures fast kinetics during charge/discharge cycling. In addition, the unique architecture of vertically aligned few-layered MoTe₂ on graphene provides more abundant exposed active edge sites for catalyzing sulfur species redox. We have included the corresponding discussion on page 5 in the revised manuscript.

Comment 3: Also the synchrotron X-ray diffraction suggests uniform sulfur distribution for MTG after 100 cycles. What about pure MT? will it be same as MTG?

Response to Comment 3: We thank the reviewer for the comment. We would have loved to test the synchrotron X-ray diffraction of MT; however, it is unfortunately shutdown for an upgrade as shown in the below screenshot of the notice posted on APS website (Figure R6 below). We tried our best to secure a reservation before the shutdown but it was hard to find one before the shut-down. We apologize for this. To verify whether the MT can lead to a uniform sulfur distribution after 100 cycles, the elemental distribution of the MT/S was obtained. As shown in Figure R7 below, MT/S shows a uniform sulfur distribution after 100 cycles, which is similar to that of MTG/S. Figure R7 has been added as Figure S5 in the revised supporting information. We have included the corresponding discussion on page 15 in the revised manuscript.

Figure R6. The notice for APS upgrade.

Figure R7. SEM image of (a) MT/S and (b) MTG/S and their corresponding elemental mapping images of sulfur after 100 cycles.

Comment 4: *Usually cathodes are prepared with pvdf binder in case of metal-sulfur batteries. Is there any particular reason to use CMC as binder for MoTe₂ instead of pvdf? What will be the performance with pvdf compared to cmc?*

Response to Comment 4: We thank the reviewer for raising this important question. The main reason why CMC was used as a binder for the MTG/S electrode is that the Na ions present in the CMC could potentially provide some ionic conductivity, which can greatly help improve the reversible capacity of Na-S batteries. In addition, compared with PVDF, the electrode with CMC as binder shows improved electrochemical performance, which has been proved in previous work [Nat. Commun. 2018, 9(1): 3870]. We have included the corresponding discussion on page 22 in the revised manuscript.

REVIEWER COMMENTS

Reviewer #1 (Remarks to the Author):

All of my concerns have been fully addressed. I recommend to publishing

Reviewer #3 (Remarks to the Author):

The science is interesting, but there I have concerns about the study. The authors claim that every little difference between their system and the control is extremely significant, but to me it seems to be making mountains out of molehills. This study provides some insight into combined intercalation-conversion electrode materials for NaS batteries, but I do not believe this paper belongs in a journal as prestigious as Nature Communications. A few specific comments:

1. The writing needs significant improvement. The language feels stiff, even for a scientific report, making it extremely difficult to read.
2. Looking at their results, the sulfur utilization and capacity fade look much improved, but given the mass difference between carbon and MoTe₂, I would like to see a comparison of the gravimetric energy density for the total mass of electrode material, not just active sulfur material.
3. The authors conclude that their work "pave[s] the way for the practical application of Na-S batteries," but I question the use of Mo and Te as practical materials. This brings into question the relevance of this study.
4. Delineation of the sulfur transformation and interfacial chemistry might be out of order. Figure 4 is showing sulfur distribution after 100 cycles without talking about the cycling performance first.
5. The voltage window is large. Can the authors comment on the practicality of a 0.8V-2.8V window?

Line notes:

40 - References for work on NaS for preventing PS shuttle. Three are from 2016 and one is from 2014. Please find more recent references. Review articles are always good resources for background.

95 - "... the utilization of active material was significantly enhanced..."...as evidenced by what?

63 - MoTe₂ Interlayer distance = 0.392nm

111 - MoTe₂ bulk interlayer spacing = 0.69nm

There seems to be some disagreement regarding the interlayer distance?

132 - Even when referring to previous work, you should quickly detail what electrolyte you are using.

142 (and Fig 3) - It is unclear how you are choosing where to mark the "plateaus" to calculate a voltage hysteresis. There are no plateaus here, it is simply a curved voltage profile. How can you provide any certain analysis when there is no clear way to interpret these data?

147-148 - Please fix this sentence fragment.

155-157 - This analogy is irrelevant and should be removed.

159 - What is the sweep rate for the CV? It is not detailed in the methods, either. How can you state that kinetics are faster in one system when the difference of a couple of mV might just be a result of your sweep rate?

166 - "attractive" -- The author may want to reconsider their word choice here.

REVIEWER 1

General Comment: All of my concerns have been fully addressed. I recommend to publishing

Response to General Comment: We greatly appreciate the reviewer for the positive feedback.

REVIEWER 2

General Comment: The science is interesting, but there I have concerns about the study. The authors claim that every little difference between their system and the control is extremely significant, but to me it seems to be making mountains out of molehills. This study provides some insight into combined intercalation-conversion electrode materials for NaS batteries, but I do not believe this paper belongs in a journal as prestigious as Nature Communications.

Response to General Comment: We appreciate the reviewer very much for the positive comments and valuable comments/suggestions, which are very useful for us to improve our manuscript. We feel sorry that we did not state so clearly the novelty of our manuscript. Here, we would like to state clearly the significant novelty of our work.

To address the challenges of Na-S batteries, intensive research efforts have been pursued, such as the combination of conductive carbon and sulfur, modification of separators with carbon, and development of sulfide cathodes. Low sulfur loading ($< 2 \text{ mg cm}^{-2}$) has illustrated decent electrochemical performance in these systems. However, recently reported work is mainly based on carbonate electrolytes, where a flooded electrolyte amount is required to wet the glass fiber separator thoroughly. In this regard, it is still challenging to achieve high-performance Na-S batteries under rigorous conditions, including high active material loading and lean electrolyte, which is crucial for practical Na-S batteries. A localized high-concentration electrolyte (LHCE) developed in our previous work enable a stable Na-S battery under a decent amount of electrolyte condition. However, the capacity is only 675 mA h g^{-1} after 300 cycles with a low sulfur loading of 2 mg cm^{-2} at a low rate of 0.1C, which is far from the theoretical capacity of $1,675 \text{ mA h g}^{-1}$ of Na-S batteries. These results indicate that the utilization of sulfur and the kinetics of the redox need to be further improved.

Herein, for the first time, we introduce a new concept of intercalation-conversion hybrid cathode to dramatically promote the kinetics of sulfur redox and improve the utilization of sulfur by coupling the intercalation-type catalyst MoTe_2 with the conversion-type active material sulfur in Na-S batteries. In addition, MoTe_2 nanosheets vertically grown on graphene flakes offer abundant active catalytic sites, further boosting the catalytic activity for sulfur redox. With the intercalation-conversion strategy and well-designed structure, a high capacity of $1,081 \text{ mA h g}^{-1}$ with a capacity fade rate of only 0.05% per cycle over 350 is realized under high sulfur loading and lean electrolyte condition. In addition, the MTG/S can still deliver a discharge capacity of 637 mA h g^{-1} even over an impressive 900 cycles, corresponding to a capacity decay rate of 0.02% per cycle. The fundamental understanding of the electrocatalysis of MoTe_2 is further revealed by *in-situ* synchrotron-based operando energy dispersive X-ray diffraction and *ex-situ* time-of-flight secondary ion mass spectrometry, which provides new insights and opportunities to develop advanced Na-S batteries with

highly efficient electrocatalysts for sulfur conversion.

The concept of intercalation-conversion strategy provides a new route for designing novel sulfur cathode for Na-S batteries. Such a concept guides further exploration of promising sulfur cathodes with intercalation-type catalysts. These catalysts will enable high-energy cathodes for developing next-generation Na-S batteries as well as Li-S batteries.

Also, the fundamental understanding of MTG in Na-S batteries and the deep discussion on the electrochemical results collected under practically necessary conditions in this manuscript can provide new insights and opportunities to develop high-performance Na-S batteries. We believe our work can motivate the rechargeable battery community to broaden their interest and benefit the energy storage area.

Comment 1: The writing needs significant improvement. The language feels stiff, even for a scientific report, making it extremely difficult to read.

Response to Comment 1: We thank the reviewer for the suggestion, which helps to improve the quality of our manuscript. As suggested, the language has been carefully improved in the revised manuscript.

Comment 2: Looking at their results, the sulfur utilization and capacity fade look much improved, but given the mass difference between carbon and MoTe₂, I would like to see a comparison of the gravimetric energy density for the total mass of electrode material, not just active sulfur material.

Response to Comment 2: We thank the reviewer for the comment. The content of the sulfur in each cathode is 70%. Here is the detailed electrode preparation:

The MTG or Ketjen Black carbon was mixed with an appropriate amount of sulfur, and the mixture was heated at 155 °C for 12 h in a sealed vial under an Ar atmosphere. Then, the as-obtained product was heated at 200 °C for 30 min in a flowing Ar atmosphere in a tube furnace to remove the redundant sulfur outside of the MTG or Ketjen Black carbon. **The sulfur content in the composite was calculated to be ~ 87 wt.%.**

For the standard Na-S batteries, **each electrode contained 80 wt.% sulfur composite**, 5 wt.% Super-P, 5 wt.% multi-walled carbon nanotubes, and 10 wt.% sodium carboxymethyl cellulose (CMC) binder.

We have added the cycling performance based on the total mass of electrode materials, which is marked with a unit of mAh g⁻¹_e, as Figure S4 in the revised supporting information. The subscript e in mAh g⁻¹_e means electrode.

Figure R1 | Cycling performances of MTG/S cathode and C/S cathode at 0.1C rate. Here, as the sulfur content in each cathode is 70%, the capacity based on the total mass of electrode materials equals 70% of the capacity based on sulfur mass. The subscript e in mAh g⁻¹_e means electrode.

Comment 3: *The authors conclude that their work "pave[s] the way for the practical application of Na-S batteries," but I question the use of Mo and Te as practical materials. This brings into question the relevance of this study.*

Response to Comment 3: We thank the reviewer for the comment. In this work, the MoTe₂ was only selected as a model system to prove the concept of the intercalation-conversion hybrid cathode to promote the kinetics of sulfur redox and improve the utilization of sulfur for Na-S batteries. As demonstrated by the electrochemical performance, the concept of intercalation-conversion hybrid cathode was well proved, which can pave the way for the practical application of Na-S batteries.

To address the concern on the practical issue, we have revised all the sentences on practical application.

Comment 4: *Delineation of the sulfur transformation and interfacial chemistry might be out of order. Figure 4 is showing sulfur distribution after 100 cycles without talking about the cycling performance first.*

Response to Comment 4: We thank the reviewer for the valuable suggestion. We moved the "Delineation of the sulfur transformation and interfacial chemistry" after the "Electrochemical performance discussion."

Comment 5: *The voltage window is large. Can the authors comment on the practicality of a 0.8V-2.8V window?*

Response to Comment 5: We thank the reviewer for the valuable suggestion. In our work, even in the range of 1.2 – 2.8V, the MTG/S can still deliver a capacity of ~700 mAh g⁻¹, which can be employed in many systems. Also, the voltage can be increased by a series connection.

To address the concern on the practical issue, we have revised all the sentences on practical application.

Comment 6: 40 - *References for work on NaS for preventing PS shuttle. Three are from 2016 and one is from 2014. Please find more recent references. Review articles are always good resources for background.*

Response to Comment 6: We thank the reviewer for the suggestion, which helps to improve the quality of our manuscript. As suggested, we have added the following more recent references in the revised manuscript.

23. Zhang, E. et al. Single-Atom Yttrium Engineering Janus Electrode for Rechargeable Na-S Batteries. *J. Am. Chem. Soc.* **144**: 18995-19007 (2022).

24. Zhou, Xue. et al. A High-Efficiency Mo₂C Electrocatalyst Promoting the Polysulfide Redox Kinetics for Na-S Batteries. *Adv. Mater.* **34**: 2200479 (2022).

25. Yang, H. et al. Architecting Freestanding Sulfur Cathodes for Superior Room-Temperature Na-S Batteries. *Adv. Funct. Mater.* **31**: 2102280 (2021).

26. Fang, D. et al. Low-Coordinated Zn-N₂ Sites as Bidirectional Atomic Catalysis for Room-Temperature Na-S Batteries. *ACS Appl. Mater. Interfaces.* **15**: 26650-26659 (2023).

Comment 7: 95 - *"... the utilization of active material was significantly enhanced..."...as evidenced by what?*

Response to Comment 7: We thank the reviewer for the valuable suggestion. We have deleted such a sentence to clear up any confusion.

Comment 8: 63 - *MoTe₂ Interlayer distance = 0.392nm*

111 - MoTe₂ bulk interlayer spacing = 0.69nm

There seems to be some disagreement regarding the interlayer distance?

Response to Comment 8: We thank the reviewer for the suggestion, which helps to improve the quality of our manuscript. According to the reference [Chemical Engineering Journal 452 (2023): 139111], as a two-dimensional family member of molybdenum-based materials, molybdenum telluride (MoTe₂) has been considered a promising electrode material for sodium-ion storage because of much wider interlayer distance of 0.699 nm, which is larger than that in traditional graphite (0.335 nm) and MoS₂ (0.615 nm) as well as MoSe₂ (0.646 nm).

This corresponding part has been modified on page 3 in the revised manuscript to maintain the consistency in the article.

Comment 9: 132 - *Even when referring to previous work, you should quickly detail what electrolyte you are using.*

Response to Comment 9: We thank the reviewer for the suggestion, which helps to improve the quality of our manuscript. As suggested, the details of LHCE electrolyte have been added on page 7 in the revised manuscript.

In detail, the LHCE consists of sodium bis(fluorosulfonyl)imide (NaFSI), 1,2-Dimethoxyethane (DME), and 1,1,2,2-tetrafluoroethyl 2,2,3,3-tetrafluoropropyl ether (TTE) with an molar ratio of 1 : 1.2 : 1.

Comment 10: 142 (and Fig 3) - It is unclear how you are choosing where to mark the "plateaus" to calculate a voltage hysteresis. There are no plateaus here, it is simply a curved voltage profile. How can you provide any certain analysis when there is no clear way to interpret these data?

Response to Comment 10: We thank the reviewer for the suggestion, which helps to improve the quality of our manuscript. To provide a clear way to calculate a voltage hysteresis (ΔE), we define ΔE as voltage difference between the charge and discharge profiles at the 50% capacity of the cells. Through this definition, the data can be clearly interpreted. We have added this definition on page 8 in the revised manuscript.

Comment 11: 147-148 - Please fix this sentence fragment.

Response to Comment 11: We thank the reviewer for the suggestion, which helps to improve the quality of our manuscript. As suggested, the sentence fragment in the revised manuscript has been fixed on page 8.

Also, the MTG/S cathode could exhibit a low voltage hysteresis of 0.73 V even in the case of high sulfur loading and low electrolyte dosage.

Comment 12: 155-157 - This analogy is irrelevant and should be removed.

Response to Comment 12: We thank the reviewer for the suggestion, which helps to improve the quality of our manuscript. As suggested, the corresponding analogy has been removed.

Comment 13: 159 - What is the sweep rate for the CV? It is not detailed in the methods, either. How can you state that kinetics are faster in one system when the difference of a couple of mV might just be a result of your sweep rate?

Response to Comment 13: We thank the reviewer for the suggestion, which helps to improve the quality of our manuscript. The CVs were recorded at the same scan rate of 0.1 mV s⁻¹, as shown in Figure 3c and Figure 3d. As suggested, the sweep rate for the CV has been added on page 8 in the revised manuscript.

Comment 14: 166 - "attractive" -- The author may want to reconsider their word choice here.

Response to Comment 14: We thank the reviewer for the suggestion, which helps to improve the quality of our manuscript. As suggested, the corresponding word has been replaced on page 9 in the revised manuscript as follows:

Additionally, a new cathodic peak at 0.89 V and an anodic peak at 1.06 V appear in the CV curve of MTG/S cathode, which are related, respectively, to the sodiation of MoTe_2 to Na_xMoTe_2 and desodiation of Na_xMoTe_2 to MoTe_2 .